# Economic Burden of Pneumococcal Disease in Individuals Aged 15 Years and Older in the Liguria Region of Italy

**DOI:** 10.3390/vaccines9121380

**Published:** 2021-11-24

**Authors:** Matteo Astengo, Chiara Paganino, Daniela Amicizia, Cecilia Trucchi, Federico Tassinari, Camilla Sticchi, Laura Sticchi, Andrea Orsi, Giancarlo Icardi, Maria Francesca Piazza, Bruno Di Silverio, Arijita Deb, Francesca Senese, Gian Marco Prandi, Filippo Ansaldi

**Affiliations:** 1Regional Health Agency of Liguria (ALiSa), 16121 Genoa, Italy; Chiara.paganino@regione.liguria.it (C.P.); Daniela.amicizia@unige.it (D.A.); Cecilia.trucchi@regione.liguria.it (C.T.); fedetassi@ymail.com (F.T.); Camilla.sticchi@regione.liguria.it (C.S.); mariafrancesca.piazza@regione.liguria.it (M.F.P.); bruno.disilverio@regione.liguria.it (B.D.S.); filippo.ansaldi@unige.it (F.A.); 2Department of Health Sciences (DiSSal), University of Genoa, 16132 Genoa, Italy; Laura.sticchi@unige.it (L.S.); Andrea.orsi@unige.it (A.O.); icardi@unige.it (G.I.); 3Merck & Co., Inc., Kenilworth, NJ 07033, USA; arijitadeb88@gmail.com; 4MSD Italy, 00189 Rome, Italy; Francesca.senese@merck.com (F.S.); gian.marco.prandi@merck.com (G.M.P.)

**Keywords:** pneumonia, pneumococcal disease, pneumococcal infections, health resources, comorbidity, healthcare costs, economic burden

## Abstract

Despite the availability of vaccines against *Streptococcus pneumoniae*, the global incidence and economic cost of pneumococcal disease (PD) among adults is still high. This retrospective cohort analysis estimated the cost of emergency department (ED) visits/hospitalizations associated with non-invasive pneumonia and invasive pneumococcal disease among individuals ≥15 years of age in the Liguria region of Italy during 2012–2018. Data from the Liguria Region Administrative Health Databases and the Ligurian Chronic Condition Data Warehouse were used, including hospital admission date, length of stay, discharge date, outpatient visits, and laboratory/imaging procedures. A ≥30-day gap between two events defined a new episode, and patients with ≥1 ED or inpatient claim for PD were identified. The total mean annual number of hospitalizations for PD was 13,450, costing ~€49 million per year. Pneumonia accounted for the majority of hospitalization costs. The median annual cost of hospitalization for all-cause pneumonia was €38,416,440 (per-capita cost: €26.78) and was €30,353,928 (per-capita cost: €20.88) for pneumococcal and unspecified pneumonia. The total number and associated costs of ED visits/hospitalizations generally increased over the study period. PD still incurs high economic costs in adults in the Liguria region of Italy.

## 1. Introduction

Pneumococcal disease (PD) is caused by *Streptococcus pneumoniae* and includes invasive (e.g., meningitis, bacteremia, sepsis, and bacteremic pneumonia) and non-invasive (e.g., sinusitis, bronchitis, and non-bacteremic pneumonia) diseases [1,2].

Invasive disease occurs when pneumococcus enters sterile sites, such as blood or cerebrospinal fluid, causing very serious conditions, such as bacteraemic pneumonia, meningitis, and sepsis. Infection from the lungs may spread to surrounding tissues, resulting in pleurisy or pericarditis. Furthermore, pneumococcus can pass from the airways to the blood, resulting in septicemia. Through the blood, the pneumococcus can reach any organ, including the meninges (meningitis), the joints (arthritis, osteomyelitis), and the peritoneum (peritonitis) [1,2].

Invasive diseases are serious clinical forms that can be life threatening. Pneumococcus is estimated to cause approximately 1.6 million deaths worldwide each year.

Specifically, overall mortality from pneumococcal bacteremia ranges between 15% and 20% in the antibiotic era. Pneumococcal endocarditis is rare but usually affects one or both left-sided valves (more often on the aortic than the mitral valve) and causes high mortality rates, ranging from 28% to 60%. Surgery may be required before a course of antibiotics is completed. Osler described the clinical triad of pneumococcal endocarditis, meningitis, and pneumonia (AKA Austrian’s syndrome) in 1881. The triad is now infrequent, but it still occurs.

Despite the availability of pneumococcal conjugate vaccines (PCVs) and a 23-valent pneumococcal polysaccharide vaccine (PPSV23) as preventative interventions, PD continues to cause significant morbidity and mortality worldwide [3,4], with severe PD in particular reported to be associated with high medical costs [5].

Nasopharyngeal colonization by *S. pneumoniae* leads to cross-transmission, infection, and disease, and it is a predominant source of transmission of the bacterium from children to adults. Therefore, the vaccination of pediatric populations can reduce the transmission of *S. pneumoniae*, providing indirect protection to adults [6,7,8]. It has been shown that a reduction in the rates of vaccine-serotype invasive PD (IPD) among vaccinated children is accompanied by a decrease in the rates among adults, potentially due to the reduction of vaccine-serotype colonization in both age groups [8].

To address the disease and economic burden of PD, a 7-valent PCV (PCV7) was licensed in the European Union (EU) in 2000 for use among infants and young children [9]. In 2003, a large-scale vaccination program against *S. pneumoniae* started in the Liguria region of Italy, an administrative region of more than 1.6 million inhabitants [10,11,12]. In summer 2010, PCV7 was replaced by a 13-valent PCV (PCV13) in the regional vaccination plan, and it has been widely administered since autumn 2010 [11]. Regarding pneumococcal vaccination in adults, Liguria started a large-scale program of vaccination in 2000 with PPV23 in all subjects over 64 years of age and in high-risk groups, in particular in those at major risk of acquiring pneumococcal infection and complications. Cumulative coverage rates, although not available for the Liguria region, showed in a region bordering Liguria in 2007 a PPV23 uptake that ranged from 26% to 31% in the elderly, with an approximate value of about 23% in adults with underlying risk conditions [11]. Since 2013, Liguria has implemented a new recommendation for pneumococcal immunization in adults, offering PCV13 and PPV23 in series in subjects aged between 70 and 75 years, and in individuals with specific comorbidities that are considered to place them at an increased risk of pneumococcal diseases, regardless of age.

Infants in Liguria are eligible to receive PCV13 at the ages of three, five, and 11 months [10,13]. Vaccine coverage rates in Ligurian infants have been more than 90% since 2007 [14,15,16]. However, vaccine coverage rates in older adults were reported to be suboptimal (~23%) [11].

Underestimation of the burden of PD is a well-known phenomenon [17,18]. Therefore, to demonstrate the potential value of higher-valent vaccines, it is important to quantify the residual and vaccine-preventable burden of invasive and non-invasive PD following the introduction of PCV7 and PCV13 into the childhood immunization schedule. Although data on the burden of PD in Italy are available, some measures of burden stratified by age and risk factors for severe outcomes, such as hospitalizations and emergency department (ED) visits, are lacking [19,20]. In addition, there is little information about direct costs among individuals with PD in Italy. These data are key for implementing preventative strategies and efficient healthcare pathways.

The aim of this retrospective cohort analysis was to estimate the cost of ED visits and hospitalizations associated with all-cause, pneumococcal, and unspecified pneumonia, bacteremia, meningitis, and acute otitis media (AOM) among individuals ≥15 years of age in the Liguria region of Italy during 2012–2018.

## 2. Materials and Methods

### 2.1. Study Design and Population

This was a retrospective observational cohort study to assess the epidemiologic and economic burden of all-cause pneumonia, unspecified or pneumococcal pneumonia, IPD, and AOM in Ligurian individuals aged ≥15 years (average population ≥15 years: 1,475,242; average total population: 1,592,058) during the period between October 2012 to September 2018.

Patients with one or more inpatient or ED claims for PD and all-cause pneumonia (based on International Classification of Diseases, Ninth Revision, Clinical Modification [ICD-9-CM] diagnosis codes; Appendix A) were identified during the study period. An interval of ≥30 days between two hospital visits, including back-to-back visits between 2 years, or in the ED defined a new episode for all case histories.

### 2.2. Data Sources

The study used data from the Liguria Region Administrative Health Databases and the Ligurian Chronic Condition Data Warehouse (CCDWH). The Ligurian Administrative Health Database stores healthcare data on the entire population of Liguria and includes data on all ED visits, outpatient visits, outpatient pharmaceutical dispensing records, and hospitalizations. The data collected include the date of visit or admission, information on diagnoses and procedures performed, length of stay, date of discharge, and discharge status (i.e., died or discharged alive). The CCDWH integrates four main routine administrative healthcare data flows belonging to the regional health service. The CCDWH records data gathered from multiple regional health service data sources (hospital discharge records (DRs), pharmaceutics, medical fee exemptions, outpatient visits, and laboratory/imaging procedures) within a specified period by means of a pre-defined algorithm based on the codes assigned to specific diagnoses and procedures [21]. Through a record-linkage system based on a civil registry database, residents’ histories of healthcare events are constructed in order to depict the chronic condition of each patient. All data are archived in a relational database by means of big-data logic [21].

### 2.3. Cost Inputs

The cost of hospitalizations was based on diagnostic-related group costs for hospital admissions [22]. Cost estimates for hospitalizations and ED access, including admission for short-term observation, were determined from the perspective of the government as payer, and were based on data from the diagnosis-related group (DRG) system and procedures performed in the ED. In accordance with the regional reimbursement system, costs related to ED access followed by hospitalization were estimated based on the DRG system only. The number of pharmaceutical prescriptions, days of drug therapy use, and associated costs were recorded during a 6-month follow-up period.

In the economic impact analysis for pneumonia, IPD, or AOM possibly caused by *S. pneumoniae*, in subjects presenting with more than one diagnosis during hospitalization, the cost was attributed to the principal diagnosis present on the discharge form.

### 2.4. Data Analyses

ED access and hospitalizations for all-cause pneumonia and pneumococcal or unspecified pneumonia, AOM, and IPD were stratified by age group (15–64 years and ≥65 years) and by underlying comorbidities.

For each outcome, the prevalence of each comorbidity was calculated per age group for the entire study period, and the 95% confidence interval (CI) range was estimated using the z-value of normal distribution method. Data were analyzed using the JMP version 13.0.0 software (SAS Institute, Cary, NC, USA). Quantitative outcome variables were summarized as mean or median and standard deviation or interquartile range, and categorical variables as frequency distributions and 95% CIs.

The mean and per capita costs of ED access, including admission for short-term observation, whether followed by hospitalization or not, were estimated for the entire observation period. Finally, a linear regression was applied to assess trends in the total costs for ED access and hospitalizations for pneumococcal or unspecified pneumonia, IPD, and AOM during the study period. A value of *p* < 0.05 was considered statistically significant.

## 3. Results

### 3.1. Patient Demographics

In total, 91,654 records of patients ≥15 years of age in the Liguria region were included in the study during the period 2012–2018. Demographics based on patient records by indication are presented in Table 1.

The prevalence of at least one comorbidity in the general Liguria population ≥ 15 years of age during the study period was 24.47% (95% CI: 24.40–24.54); cardiovascular disease in 8.45% (95% CI: 8.35–8.56) of patients, neoplasia in 5.91% (95% CI: 5.83–6.00), respiratory disease in 3.90% (95% CI: 3.83–3.97), diabetes in 5.49% (95% CI: 5.40–5.57), and chronic renal failure in 1.26% (95% CI: 1.22–1.30).

In the study population, 75.03% (95% CI: 74.75–75.31) of patients had at least one comorbidity; cardiovascular disease in 48.83% (95% CI: 48.5–49.15), neoplasia in 18.76% (95% CI: 18.51–19.01), respiratory disease in 23.72% (95% CI: 23.44–23.99), diabetes in 18.02% (95% CI: 17.77–18.27), chronic renal failure in 15.01% (95% CI: 14.78–15.24), and other in 33.38% (95% CI: 33.08–33.69).

### 3.2. Total Number and Cost of ED Visits and Hospitalizations

The median annual number of ED visits and hospitalizations as a result of pneumococcal and unspecified pneumonia (excluding all-cause pneumonia), IPD, and AOM was 13,450, with a median annual cost of €50.63 million, in individuals aged ≥15 years during 2012–2018. Total costs increased by 37% over the study period (€41.15 million in 2012 and €56.50 million in 2018), the cost per ED visit/hospitalization increased by 6% over the study period (€3454 in 2012 and €3671 in 2018), and the annual cost per capita increased by 42% (€28.28 in 2012 and €40.17 in 2018). Linear regression results showed a statistically significant upward trend in total costs for pneumococcal or unspecified pneumonia, IPD, and AOM during the study period (*p* < 0.0001).

### 3.3. Number and Cost of ED Visits and Hospitalizations for Pneumonia, Meningitis, and Bacteremia

The number and cost of ED visits and hospitalizations for all-cause pneumonia, pneumococcal and unspecified pneumonia, meningitis, and bacteremia are shown in Table 2, Table 3, Table 4 and Table 5. Pneumonia accounted for the majority of hospitalization costs (Table 2 and Table 3). The median annual cost of hospitalization for all-cause pneumonia and pneumococcal and unspecified pneumonia amounted to €38,416,440 (per-capita cost: €26.78) and €30,353,928 (per-capita cost: €20.88), respectively. Pneumococcal and unspecified pneumonia constituted 60% of total costs for hospitalization due to pneumococcal or unspecified diseases. Total access to ED visits and hospitalizations increased over the study period due to all-cause pneumonia (8324 in 2012 vs. 12,472 in 2018), pneumococcal-specific pneumonia (79 vs. 169), and unspecified pneumonia (6802 vs. 10,122) (Table 2 and Table 3). There was also up to an approximately 50% increase in costs of ED visits and hospitalizations due to pneumonia during the study (Table 2 and Table 3).

On the other hand, meningitis accounted for a relatively low number of ED visits and hospitalizations during the study period (Table 4). The number of ED visits and hospitalizations due to pneumococcal meningitis varied between 10 and 23 per year, costing between €5529 and €13,016 per event. For unspecified meningitis, there were between five and 11 ED visits and hospitalizations per year, costing between €6323 and €15,669 per event. There were only two ED visits due to unspecified meningitis not leading to hospitalization, in 2013 and 2015, costing €300 and €61, respectively.

The number of cases and costs of bacteremia increased over the study period (Table 5). The number of ED visits and hospitalizations for pneumococcal bacteremia increased from 85 events in 2012 to 149 events in 2018. The cost per event also increased from €6476 to €8395 during the study period. A similar pattern was seen with unspecified bacteremia. Only six ED visits due to unspecified bacteremia not followed by hospitalization were recorded in patients >64 years of age, in 2012, 2017, and 2018, costing a total of €1533 during the study period.

### 3.4. Prescribed Medication during Follow-Up

The number of pharmaceutical prescriptions, days of drug therapy, and associated costs are shown in Table 6. The median annual number of pharmaceutical prescriptions in the follow-up period was 13,541, with a median annual cost of €7,997,000, including €340,140 for antibacterial agents.

## 4. Discussion

This study has shown that, despite routine vaccination of children, there are still substantial economic costs associated with all-cause pneumonia, pneumococcal and unspecified pneumonia, bacteremia, meningitis, and AOM in individuals ≥15 years of age in the Liguria region of Italy. The Ligurian infant immunization program has achieved high levels of vaccine coverage (>90%) since 2007. However, this study shows that the economic impact of PD on adults remains high, with an annual cost of €48.97 million during the study period. Furthermore, immunization programs addressed to adult high-risk populations and the elderly are also implemented with unsatisfactory vaccination coverages.

In our study, all-cause pneumonia and pneumococcal and unspecified pneumonia accounted for the majority of hospitalization costs during the study. The median annual cost of hospitalization for all-cause pneumonia was €38,416,440 (per-capita cost: €26.78) and €30,353,928 (per-capita cost: €20.88) for pneumococcal pneumonia and unspecified pneumonia. Bacteremia and meningitis accounted for a lower proportion of costs of ED visits and hospitalizations. The costs associated with ED visits and hospitalizations due to PD also generally increased during the study period from 2012 to 2018.

A substantial healthcare utilization cost due to PD was reported in Italy as a whole, as well as in other countries. In Italy, costs related to PD have increased regardless of the introduction of PCVs in infants. In Italian adults >64 years of age, hospitalization costs for PD increased by 33.8% after the introduction of PCV7/PCV13 [23]. In Spain, the estimated cost of hospitalizations in 2011 in adults ≥18 years of age was more than €57 million for all PD, and €47 million for pneumococcal pneumonia [24]. In the United States in 2015, total costs associated with PD in adults ≥19 years of age were US$1.86 billion, of which US$1.8 billion was attributable to direct inpatient, outpatient, and medication costs [25]. PD-related costs are higher in adults with high-risk conditions and in those with IPD [24,26], and our study certainly reports a higher level of comorbidities in the study population compared with the general Ligurian adult population.

A key strength of the study is that it assessed episodes of PD as well as disease of unknown etiology. As such, it provided both a highly specific estimate of cost, with a high degree of certainty that episodes included in the analysis were caused by *S. pneumoniae*, and one that is more sensitive, with the potential to capture additional undetected pneumococcal episodes and those potentially caused by other pathogens. However, there are also some methodological limitations that must be considered. Firstly, the presence of a diagnosis code does not necessarily indicate the presence of disease, due to the possibility of incorrect coding.

In fact, analysing a total of 70,402 records occurring in Liguria from 2012–2018 with at least one ICD-9 CM diagnosis code suggestive of all-cause pneumonia, only 1.37% hospitalizations had specific ICD-9 CM diagnosis codes for pneumococcal pneumonia. This apparent low impact can be related to the available data on hospital discharge records in Italy, which are subject to some limitations (common to all passive surveillance systems), such as underestimations or deficiencies in reporting, due especially to the frequent lack of etiological identification of the causal agent during the medical care of the disease.

As a consequence, the etiologic fraction attributable to *S. pneumoniae* may be under-diagnosed or, alternatively, overestimated, considering all potentially attributable pneumococcal diseases in the absence of microbiological identification.

Accordingly, several studies have reported that ICD-9 CM codes suggest that pneumococcal pneumonia have good specificity, but poor sensitivity [27,28]. Thus, our results represent a good starting point to reinforce strategies addressed to improve more sensitive surveillance systems that are needed nationwide. Hospitals in particular, where more severe cases are observed, could be one of the key places where a specific surveillance program can be performed with the most updated techniques of molecular biology.

Secondly, indirect costs, such as productivity loss, the cost of transportation, and loss of earnings, were not taken into account.

Finally, in our study, we used ICD-9-CM codification, although the standard is now at its eleventh revision (ICD-11), approved in 2018 by the Member States, at the 72 World Health Assembly. The decision to use the old coding system could suggest some inaccuracies in our data, but in Italy the new version mentioned above will come into force on 1 January 2022.

ICD-9-CM is the version of ICD currently utilised in Italy, according to legislation, for the coding of diseases and related problems, for statistical studies on morbidity and mortality rates, and for epidemiological studies. It is also recognized as a valid management tool for public health and hygiene [29].

## 5. Conclusions

Despite widespread pediatric and older adult vaccination programs in Liguria, PD in adults continues to incur high economic costs due to ED visits and hospitalizations. In this study, the majority of costs were due to all-cause, pneumococcal-specific, and unspecified pneumonia. The results of this study confirm the need to reinforce vaccination programs in adults at risk, especially those with comorbidities.

## Figures and Tables

**Table 1 vaccines-09-01380-t001:** Demographics of patient records.

	All-Cause Pneumonia (*n* = 70,402)	Pneumonia (*n* = 61,559)		Meningitis (*n* = 192)		Bacteremia (*n* = 8587)		Otitis Media (*n* = 11,061)	Other Invasive Diseases * (*n* = 7463)	Total (*n* = 91,654)
		Pneumococcal (*n* = 967)	Unspecified (*n* = 60,592)	Pneumococcal (*n* = 129)	Unspecified (*n* = 63)	Pneumococcal (*n* = 121)	Unspecified (*n* = 8466)			
Male, *n*; % [95% CI]	37,830; 53.73 [53.37–54.10]	513; 53.05 [49.90–56.19]	32,380; 53.43 [53.04–53.83]	63; 48.83 [40.21–57.46]	27; 42.85 [30.63–55.07]	65; 53.71 [44.83–62.60]	4440; 52.44 [51.38–53.50]	5640; 50.98 [50.05–51.92]	5640; 75.57 [74.6–76.55]	49,168; 53.65 [53.32–53.97]
Mean age (SD)	72.8 (18.76)	73.20 (16.63)	75.23 (16.83)	62.55 (14.89)	63.29 (17.82)	70.53 (16.06)	75.69 (14.72)	46.58 (18.39)	66.26 (15.92)	69.56 (20.28)
Age group, *n*; % [95% CI]										
15–64 years	18,077; 25.68 [25.35–26.00]	272; 28.13 [25.29–30.96]	13,284; 21.92 [21.59–22.25]	63; 48.84 [40.21–57.46]	26; 41.27 [29.11–53.43]	50; 41.32 [32.55–50.10]	1709; 20.19 [19.33–21.04]	9077; 82.06 [81.35–82.78]	2202; 29.51 [28.47–30.54]	29,940; 32.67 [32.36–32.97]
≥65 years	52,325; 74.32 [74.00–74.65]	695; 71.87 [69.04–74.71]	47,308; 78.08 [77.75–78.41]	66; 51.16 [42.54–59.79]	37; 58.73 [46.57–70.89]	71; 58.68 [49.90–67.45]	6757; 79.81 [78.96–80.67]	1984; 17.94 [17.22–18.65]	5261; 70.49 [69.46–71.53]	61,714; 67.33 [67.03–67.64]
Any comorbidity *n*; % [95%CI]	56,705; 80.54 [80.25–80.84]	786; 81.28 [78.82–83.74]	49,872; 82.31 [82.00–82.61]	87; 67.44 [59.36–75.53]	47; 74.60 [63.85–85.35]	93; 76.86 [69.35–84.37]	7614; 89.94 [89.30–90.58]	3168; 28.64 [27.80–29.48]	6215; 83.28 [82.43–84.12]	68,765; 75.03 [74.75–75.31]
Chronic renal failure	11,386; 16.17 [15.90–16.44]	130; 13.44 [11.29–15.59]	10,508; 17.34 [17.04–17.64]	6; 4.65 [1.02–8.29]	6; 9.52 [2.28–16.77]	14; 11.57 [5.87–17.27]	2028; 23.95 [23.05–24.86]	122; 1.10 [0.91–1.30]	1322; 17.71 [16.85–18.58]	13,760; 15.01 [14.78–15.24]
Cardiovascular disease	38,109; 54.13 [53.76–54.50]	469; 48.50 [45.35–51.65]	34,603; 57.11 [56.71–57.50]	36; 27.91 [20.17–35.65]	23; 36.51 [24.62–48.40]	45; 37.19 [28.58–45.80]	5079; 59.99 [58.95–61.04]	703; 6.36 [5.90–6.81]	4359; 58.41 [57.29–59.53]	44,752; 48.83 [48.5–49.15]
Respiratory disease	19,734; 28.03 [27.70–28.36]	255; 26.37 [23.59–29.15]	16,599; 27.39 [27.04–27.75]	10; 7.75 [3.14–12.37]	4; 6.35 [0.33–12.37]	19; 15.70 [9.22–22.19]	1441; 17.02 [16.22–17.82]	510; 4.61 [4.22–5.00]	1460; 19.56 [18.66–20.46]	21,738; 23.72 [23.44–23.99]
Diabetes	13,408; 19.04 [18.75–19.33]	184; 19.03 [16.55–21.50]	11,686; 19.29 [18.97–19.60]	16; 12.40 [6.71–18.09]	6; 9.52 [2.28–16.77]	15; 12.40 [6.52–18.27]	2040; 24.09 [23.19–25.01]	531; 4.80 [4.40–5.20]	1647; 22.07 [21.13–23.01]	16,516; 18.02 [17.77–18.27]
Neoplasia	13,759; 19.54 [19.25–19.84]	204; 21.10 [18.52–23.67]	12,096; 19.96 [19.64–20.28]	20; 15.50 [9.26–21.75]	13; 20.63 [10.64–30.63]	28; 23.14 [15.63–30.65]	2739; 32.35 [31.36–33.35]	538; 4.86 [4.46–5.26]	1433; 19.2 [18.31–20.1]	17,195; 18.76 [18.51–19.01]
Others	25,517; 36.24 [35.89–36.60]	370; 38.26 [35.20–41.33]	22,611; 37.32 [36.93–37.70]	31; 24.03 [16.66–31.40]	16; 25.40 [14.65–36.15]	39; 32.23 [23.90–40.56]	3770; 44.53 [43.47–45.59]	1029; 9.30 [8.76–9.84]	2429; 32.55 [31.48–33.61]	30,595; 33.38 [33.08–33.69]

* At least one of (ICD-9-CM code): Infective pericarditis (420.9), acute and subacute endocarditis (421.1–421.9), empyema (510.9), pneumococcal peritonitis (567.1), spontaneous bacterial peritonitis (567.23), pyogenic arthritis (711.0/711.9), acute osteomyelitis (730.0x–730.2x), pleural effusion with unspecified bacterial cause (511.1), AOM complication with miryngotomy/tympanostomy with drainage of middle ear (20.0). AOM: acute otitis media; CI: confidence interval; ICD-9-CM: International Classification of Diseases, Ninth Revision, Clinical Modification; SD, standard deviation.

**Table 2 vaccines-09-01380-t002:** Healthcare resource utilization and costs for all-cause pneumonia for subjects ≥15 years of age in the Liguria region 2012–2018.

Year	ED Visits (Not Followed by Hospitalization)			ED Visits + Other Hospitalization			Total Access to ED and Hospitalizations		
	Number	Costs (€)	Cost per Event (€)	Number	Costs (€)	Cost per Event (€)	Number	Costs (€)	Cost per Event (€)
2012	1114	104,920	94.18	7210	32,028,752	4442	8324	32,133,672	3860
2013	1210	119,713	98.94	7459	32,968,282	4420	8669	33,087,995	3817
2014	1684	169,536	100.67	7765	35,275,129	4543	9446	35,444,665	3752
2015	1681	181,105	107.74	8689	38,705,514	4455	10,370	38,886,619	3750
2016	1600	177,025	110.64	8436	38,416,440	4554	10,036	38,593,465	3846
2017	1800	203,609	113.12	9326	40,576,454	4351	11,126	40,780,063	3665
2018	2126	234,526	110.31	10,346	43,694,624	4223	12,472	43,929,150	3522
Weighted average		168,954	106.15		37,845,435	4418		38,291,363	3219

ED: emergency department.

**Table 3 vaccines-09-01380-t003:** Healthcare resource utilization and costs for specified and unspecified pneumonia for individuals ≥15 years of age in the Liguria region 2012–2018.

Year	ED Visits (Not Followed by Hospitalization)			ED Visits + Other Hospitalization			Total Access to Hospitalization and ED		
	Number	Costs (€)	Cost per Event (€)	Number	Costs (€)	Cost per Event (€)	Number	Costs (€)	Cost per Event (€)
Pneumococcal pneumonia									
2012	0	0	–	79	380,836	4821	79	380,836	4821
2013	0	0	–	121	610,601	5046	121	610,601	5046
2014	0	0	–	118	611,587	5183	118	611,587	5183
2015	0	0	–	121	531,407	4392	121	531,407	4392
2016	0	0	–	107	517,943	4841	107	517,943	4841
2017	0	0	–	158	784,849	4967	158	784,849	4967
2018	0	0	–	169	754,212	4463	169	754,212	4463
Weighted average					626,946	4801		626,946	4801
Unspecified pneumonia									
2012	173	16,931	98	6629	23,302,741	3515	6802	23,319,672	3428
2013	163	16,397	101	6925	24,237,574	3500	7088	24,253,971	3422
2014	165	18,164	110	7555	26,495,484	3507	7720	26,513,648	3434
2015	197	25,343	129	8310	29,987,075	3609	8507	30,012,418	3528
2016	226	27,615	122	8047	29,835,985	3708	8273	29,863,600	3610
2017	213	25,229	118	8974	32,355,936	3606	9187	32,381,165	3525
2018	244	27,590	113	9878	34,706,579	3514	10,122	34,734,169	3432
Weighted average		18,740	114		29,208,583	3568		29,232,052	3485

ED: emergency department.

**Table 4 vaccines-09-01380-t004:** Healthcare resource utilization and costs for meningitis for individuals ≥15 years of age in the Liguria region 2012–2018.

Year	ED Visits + Other Hospitalization				Total Access to Hospitalization and ED		
	Number	Number	Costs (€)	Cost per Event (€)	Number	Costs (€)	Cost per Event (€)
Pneumococcal meningitis							
2012	0	12	79,267	6606	12	79,267	6606
2013	0	15	147,537	9836	15	147,537	9836
2014	0	10	55,286	5529	10	55,286	5529
2015	0	20	260,327	13,016	20	260,327	13,016
2016	0	18	144,186	8010	18	144,186	8010
2017	0	23	262,049	11,393	23	262,049	11,393
2018	0	19	175,319	9227	19	175,319	9227
Weighted average			178,438	9606		178,438	9606
Unspecified meningitis							
2012	0	10	104,321	10,432	10	104,321	10,432
2013	1	4	78,047	19,512	5	78,347	15,669
2014	0	8	50,587	6323	8	50,587	6323
2015	1	10	113,323	11,332	11	113,384	10,308
2016	0	9	66,686	7410	9	66,686	7410
2017	0	6	40,988	6831	6	40,988	6831
2018	0	10	93,125	9313	10	93,125	9313
Weighted average			81,942	9598		82,444	9279

ED: emergency department.

**Table 5 vaccines-09-01380-t005:** Healthcare resource utilization and costs for bacteremia for individuals ≥15 years of age in the Liguria region 2012–2018.

Year	ED Visits + Other Hospitalization			Total Access to Hospitalization and ED		
	Number	Costs (€)	Cost per Event (€)	Number	Costs (€)	Cost per Event (€)
Pneumococcal bacteremia						
2012	85	550,487	6476	85	550,487	6476
2013	114	840,435	7372	114	840,435	7372
2014	129	783,265	6072	129	783,265	6072
2015	134	921,595	6878	134	921,595	6878
2016	166	1,291,753	7782	166	1,291,753	7782
2017	171	1,431,004	8368	171	1,431,004	8368
2018	149	1,250,853	8395	149	1,250,853	8395
Weighted average		1,068,192	7457		1,068,192	7457
Unspecified bacteremia						
2012	951	5,248,597	5519	954	5,249,230	5502
2013	979	5,202,355	5314	979	5,202,355	5314
2014	1116	6,459,316	5788	1116	6,459,316	5788
2015	1244	7,983,688	6418	1244	7,983,688	6418
2016	1252	8,627,310	6891	1252	8,627,310	6891
2017	1224	8,205,921	6704	1225	8,206,221	6699
2018	1221	8,064,550	6605	1223	8,065,150	6595
Weighted average		7,362,056	6283		7,362,261	6279

ED: emergency department; NR: not reported.

**Table 6 vaccines-09-01380-t006:** Pharmaceutical and antibiotics prescriptions costs during the 6-month follow-up period for individuals ≥15 years of age in the Liguria region 2012–2018.

Year	Pharmaceutical Prescriptions			Antimicrobials			Antibacterials		
	Number	Days of Drug Therapy	Costs (€)	Number	Days of Drug Therapy	Costs (€)	Number	Days of Drug Therapy	Costs (€)
2012	11,912	4,036,659	5,814,942	6640	131,825	713,788	6481	98,933	339,138
2013	12,206	4,225,424	6,313,103	6753	135,076	791,291	6591	103,664	372,499
2014	12,837	4,662,019	6,740,202	7361	142,900	813,195	7209	110,989	370,273
2015	13,675	5,237,878	7,996,969	7553	155,272	1,000,425	7397	118,067	330,834
2016	13,541	5,457,865	8,800,113	7457	158,361	1,522,924	7284	120,098	340,142
2017	14,583	6,265,637	8,802,385	7906	172,000	1,169,078	7743	132,478	312,532
2018	15,393	7,343,894	11,010,118	8372	171,744	1,294,140	8182	135,981	340,541
Median annual cost			7,996,969			1,000,425			340,140

## Data Availability

Merck Sharp & Dohme Corp., a subsidiary of Merck & Co., Inc., Kenilworth, NJ, USA’s data sharing policy, including restrictions, is available at http://engagezone.msd.com/ds_documentation.php (accessed on 9 November 2021) through the EngageZone site or via email to dataaccess@merck.com.

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
