# Peer review of "Economic Burden of Pneumococcal Disease in Individuals Aged 15 Years and Older in the Liguria Region of Italy"

_vaccines, 2021, doi:10.3390/vaccines9121380_

Round 1
Reviewer 1 Report
- The median values included in the text from pages 7-8 should be included also in table 6
- It will be interesting to evaluate the cost of treatment for patients with different risk factors to see if these increased the severity of the disease and the treatment cost.
- In discussion section, you can evaluate how antibiotic resistance in pneumonia patients can influence the cost of the treatment, see 10.31925/farmacia.2020.3.17
Author Response
- The median values included in the text from pages 7-8 should be included also in table 6
As required, we added median values in table 6.
- It will be interesting to evaluate the cost of treatment for patients with different risk factors to see if these increased the severity of the disease and the treatment cost.
We agree with reviewer comment, however as previously described for reviewer 1, this aspect has not been deepened in our study and this issue could be carried out in future studies.
- In discussion section, you can evaluate how antibiotic resistance in pneumonia patients can influence the cost of the treatment, see 10.31925/farmacia.2020.3.17
We agree with reviewer comment, it will be very interesting investigate how antibiotic resistance in pneumonia patients can influence the cost of the treatment, however this would require time and resources, in this emergency pandemic and could be performed in next studies.
Reviewer 2 Report
Economic Burden of Pneumococcal Disease in Individuals Aged 15 Years and Older in the Liguria Region of Italy.
Article describing the economic cost of Pneumococcal Disease in an Italian region in the era of PCVs. The article is well written and understandable.
Major comment
From my point of view, the main difficulty of this study is to assess whether the coded diagnoses are reliable or not. The proportion of Pneumococcal pneumonia (967/70,402, aprox. 1.4%) of all pneumonias seems very low. Does this mean the infra-diagnosis of pneumococcal pneumonia? Commonly, S. pneumoniae is the etiological agent found in around 20-30% CAP (Ferreira-Coimbra J, Sarda C, Rello J. Burden of Community-Acquired Pneumonia and Unmet Clinical Needs. Adv Ther. 2020;37:1302-8.). These data suggests a very deficient complementation of Diseases coding and compromise the results and conclusions of the work. On the other hand, assuming all unspecified pneumonia as pneumococcal pneumonia will oversize the burden of pneumococcal pneumonia. All this means that with the data provided, a reader cannot know the real economic impact of pneumococcal disease, which is the tittle of the work.
I think that the authors should discuss this issue (discrepancy between data expected and found on the prevalence of the infectious diseases analysed) in more detail to give the study greater veracity.
Minor comments.
I suggest to use less decimal figures when writing percentages (Cole TJ. Too many digits: the presentation of numerical data. Arch Dis Child. 2015;100:608-9.)
Introduction.
The authors mention the use of PCV13 in older adults. In other countries, the vaccine used is PPV23. Is PPV23 also used in Liguria? If yes, please point it out in the Introduction.
Material and methods.
2.1. Please, point out the total and ≥15 years-old annual average population of the Liguria region.
2.3. Cost inputs. Economic impact of S. pneumoniae pneumonia, IPD, AOM, the cost was attributed to the principal diagnosis. Considering the high proportion of comorbidities, (75%), could the authors say the % of PD cases whose cost were attributed to a diagnosis different from PD?
Results.
Table 3, Pneumococcal pneumonia. Where all cases of pneumococcal pneumonia visiting ED admitted to hospital? The column on the left (ED visits (not followed by hospitalization) show cero cases all years. Also, the figures of ED visits (not followed by hospitalization) in “all-cause pneumonia” and “Unspecified pneumonia” seems very low (around 0.4% and 2.5%, respectively).
Other reports from Italy showed hospitalization rates between 7% and 35%
(Sterrantino C, et al. Burden of community-acquired pneumonia in Italian general practice. Eur Respir J. 2013;42:1739-42.
Viegi G, et al. Epidemiological survey on incidence and treatment of community acquired pneumonia in Italy. Respir Med 2006;100:46-55.
The total of cases (n) in Table 1 does not match with total “number” cases of Tables 2 to 5 (for instance, 967 pneumococcal pneumonia in Table 1 and 873 in Table 3). In addition, total number of cases is not calculated in Tables 2-5. Please, show.
Also, the median annual cost for hospitalization for all-cause pneumonia €38,416,440 does not match with the €37,845,435 shown in Table 2. The same for pneumococcal and unspecified pneumonia.
Author Response
Economic Burden of Pneumococcal Disease in Individuals Aged 15 Years and Older in the Liguria Region of Italy.
Article describing the economic cost of Pneumococcal Disease in an Italian region in the era of PCVs. The article is well written and understandable.
Major comment
From my point of view, the main difficulty of this study is to assess whether the coded diagnoses are reliable or not. The proportion of Pneumococcal pneumonia (967/70,402, aprox. 1.4%) of all pneumonias seems very low. Does this mean the infra-diagnosis of pneumococcal pneumonia? Commonly, S. pneumoniae is the etiological agent found in around 20-30% CAP (Ferreira-Coimbra J, Sarda C, Rello J. Burden of Community-Acquired Pneumonia and Unmet Clinical Needs. Adv Ther. 2020;37:1302-8.). These data suggests a very deficient complementation of Diseases coding and compromise the results and conclusions of the work. On the other hand, assuming all unspecified pneumonia as pneumococcal pneumonia will oversize the burden of pneumococcal pneumonia. All this means that with the data provided, a reader cannot know the real economic impact of pneumococcal disease, which is the tittle of the work.
I think that the authors should discuss this issue (discrepancy between data expected and found on the prevalence of the infectious diseases analysed) in more detail to give the study greater veracity.
-Thank you for your comment, this is not only a prerogative of our study, but it represents a real nationwide issue that we try to explain in more details in the following paragraph and in the discussion section in the main text.
As suggested, analysing a total of 70,402 records occurring in Liguria from 2012 to 2018 with at least one ICD-9 CM diagnosis code suggestive of all-cause pneumonia, only 1.37% hospitalizations had specific ICD-9 CM diagnosis codes for pneumococcal pneumonia. This apparent low impact can be related to the available data on hospital discharge records in Italy, that are subject to some limitations (common to all passive surveillance systems), such as an underestimate or deficiencies in reporting, due especially to the frequent lack of etiological identification of the causal agent during the medical care of the disease.
In consideration of this, several studies have reported that ICD-9 CM codes suggestive of pneumococcal pneumonia have good specificity, but poor sensitivity [Guevara RE, Butler JC, Marston BJ, Plouffe JF, File TM Jr., Breiman RF. Accuracy of ICD-9-CM codes in detecting community-acquired pneumococcal pneumonia for incidence and vaccine efficacy studies. Am J Epidemiol 1999; 149:282-9; PMID:9927225; http://dx.doi.org/10.1093/oxfordjournals.aje.a009804 12. van de Garde EM, Oosterheert JJ, Bonten M, Kaplan RC, Leufkens HG. International classification of diseases codes showed modest sensitivity for detecting community-acquired pneumonia. J Clin Epidemiol 2007; 60:834-8; PMID:1760618].
Thus, our results represent a good starting point to reinforce strategies addressed to improve more sensitive surveillance systems that are needed nationwide, and just the hospital, where more severe cases are observed, could be one of the key places where a specific surveillance can be performed with the most updated techniques of molecular biology.
Minor comments.
I suggest to use less decimal figures when writing percentages (Cole TJ. Too many digits: the presentation of numerical data. Arch Dis Child. 2015;100:608-9.)
-As indicated in the reference we reported all percentages with 2 effective digits.
Introduction.
The authors mention the use of PCV13 in older adults. In other countries, the vaccine used is PPV23. Is PPV23 also used in Liguria? If yes, please point it out in the Introduction.
-Thank you for your suggestion; as required, we better described this point in the following paragraph and in the introduction section.
Regarding to pneumococcal vaccination in adults, in 2000 Liguria started a large-scale program of vaccination with PPV23 in all subjects over 64 years of age and in high risk groups, in particular in those at major risk of acquiring pneumococcal infection and complications. Cumulative coverage rates, although not available for Liguria Region, showed in a region bordering Liguria in 2007 a PPV23 uptake ranged from 26% to 31% in the elderly, with an approximate value about 23% in adults with underlying risk conditions [Orsi, A.; Ansaldi, F.; Trucchi, C.; Rosselli, R.; Icardi, G. Pneumococcus and the elderly in Italy: A summary of available evidence regarding carriage, clinical burden of lower respiratory tract infections and on-field effectiveness of PCV13 vaccination. Int J Mol Sci 2016, 17, 1140, doi:10.3390/ijms17071140.].
Since 2013, Liguria has implemented a new recommendation for pneumococcal immunization in adults, offering PCV13 and PPV23 in series in subjects aged between 70 and 75 years, and in individuals with specific comorbidities that are considered to place them at an increased risk of pneumococcal diseases, regardless of age [Orsi, A.; Ansaldi, F.; Durando, P.; Turello, V.; Icardi, G.; Gruppo di studio ligure sullo pneumococco. [Immunization campaign with 13-valent Pneumococcal Conjugate Vaccine in adults in Liguria Region, Italy: one year post-introduction preliminary results]. however, vaccine coverage rates in older adults were reported to be suboptimal (~23%) [Orsi, A.; Ansaldi, F.; Trucchi, C.; Rosselli, R.; Icardi, G. Pneumococcus and the elderly in Italy: A summary of available evidence regarding carriage, clinical burden of lower respiratory tract infections and on-field effectiveness of PCV13 vaccination. Int J Mol Sci 2016, 17, 1140, doi:10.3390/ijms17071140.].
Material and methods.
2.1. Please, point out the total and ≥15 years-old annual average population of the Liguria region.
-We added the total and ≥15 years-old annual average population in the “Study Design and Population” paragraph and we reported in this file the paragraph as appear in the main text.
This was a retrospective observational cohort study to assess the epidemiologic and economic burden of all-cause pneumonia, unspecified or pneumococcal pneumonia, and IPD in Ligurian individuals aged ≥15 years (average population ≥15 years: 1,475,242; average total population: 1,592,058) during the period October 2012 to September 2018.
2.3. Cost inputs. Economic impact of S. pneumoniae pneumonia, IPD, AOM, the cost was attributed to the principal diagnosis. Considering the high proportion of comorbidities, (75%), could the authors say the % of PD cases whose cost were attributed to a diagnosis different from PD?
-As reported in the main text, cost estimates for hospitalizations and ED access, including admission for short-term observation, were determined from the perspective of the government as payer, and were based on data from the diagnosis-related group (DRG) system and procedures performed in the ED.
The costs attributed to each DRG are defined on ministerial standardized tables and it is not possible to separate the % of PD cases whose cost were attributed to a diagnosis different from PD, because the cost is the result of different variables considered as a whole.
This issue is known, indeed some models have been performed to solve this criticism such as the following studies: Sohn, S., Hong, K., & Chun, B. C. (2020). Evaluation of the effectiveness of pneumococcal conjugate vaccine for children in Korea with high vaccine coverage using a propensity score matched national population cohort. International journal of infectious diseases: IJID : official publication of the International Society for Infectious Diseases, 93, 146–150. https://doi.org/10.1016/j.ijid.2020.01.034; Glynn RJ, Schneeweiss S, Sturmer T. Indications for propensity scores and review of their use in pharmacoepidemiology. Basic Clin Pharmacol Toxicol 2006;98 (3):253–9.
Weycker D, Strutton D, Edelsberg J, et al. Clinical and economic burden of pneumococcal disease in older US adults. Vaccine. 2010;28:4955–60. doi: 10.1016/j.vaccine.2010.05.030.
However, this aspect has not been deepened in our study.
Results.
Table 3, Pneumococcal pneumonia. Where all cases of pneumococcal pneumonia visiting ED admitted to hospital? The column on the left (ED visits (not followed by hospitalization) show cero cases all years. Also, the figures of ED visits (not followed by hospitalization) in “all-cause pneumonia” and “Unspecified pneumonia” seems very low (around 0.4% and 2.5%, respectively).
Other reports from Italy showed hospitalization rates between 7% and 35%
(Sterrantino C, et al. Burden of community-acquired pneumonia in Italian general practice. Eur Respir J. 2013;42:1739-42.
Viegi G, et al. Epidemiological survey on incidence and treatment of community acquired pneumonia in Italy. Respir Med 2006;100:46-55.
-As already reported in the comment of the reviewer 1 the low number of pneumococcal pneumonia can be related to the available data on hospital discharge records in Italy, that are subject to some limitations (common to all passive surveillance systems in Italy), such as an underestimation or deficiencies in reporting, especially due to the frequent lack of etiological identification of the causal agent during the medical care of the disease.
Furthermore, the scientific references that you recommended us are based on community-acquired pneumonia not distinguished for pneumococcal pneumonia and collected with different ways from ours with different case definitions. In fact as also admitted by authors of “Sterrantino C, et al. Burden of community-acquired pneumonia in Italian general practice. Eur Respir J. 2013;42:1739-42”, the CAP estimates are higher than those reported in adults from other studies conducted in southern Europe that can be explained by different definitions of CAP. In the limitations of the study authors reported that they may have partially overestimated the incidence of CAP due to misclassification in the algorithm coding use.
Finally, also Viegi et al. based their study on community-acquired pneumonia incidences, but not specific for pneumococcal pneumonia. Data collection is, also in this case, different from our study because they collected data starting from family practioners, implementing a modality that obviously favour the sample increase.
The total of cases (n) in Table 1 does not match with total “number” cases of Tables 2 to 5 (for instance, 967 pneumococcal pneumonia in Table 1 and 873 in Table 3). In addition, total number of cases is not calculated in Tables 2-5. Please, show.
-The cost is not attributed to more events, but to a single hospital admission. For this reason there are some discrepancies from table 1 respect to other tables. For example, one hospital admission with cost x is referred to two or more different incident events, but we attributed the cost to the principal diagnosis. In this way, in order to evaluate the economic impact of PD, we did not consider twice the cost of the same hospitalization.
Also, the median annual cost for hospitalization for all-cause pneumonia €38,416,440 does not match with the €37,845,435 shown in Table 2. The same for pneumococcal and unspecified pneumonia.
-As described in the main text, €38,416,440 is a median annual cost, while €37,845,435 is a weighted average on hospital admissions.
Reviewer 3 Report
The manuscript is generally well written. Attention to the following should improve it.
Minor comments:
- How should one differentiate between sepsis, bacteremia, and bacteremic pneumonia? Please clarify.
- The introduction would have more impact if you added the following:
a) Overall mortality from pneumococcal bacteremia ranges between 15 to 20 % in the antibiotic era. Pneumococcal endocarditis is rare but usually affects one or both left-sided valves (aortic > mitral). It’s important, because mortality remains high, ranging from 28–60%. Surgery may be required before a course of antibiotics is completed.
b) Osler described the clinical triad of pneumococcal endocarditis, meningitis, and pneumonia (AKA Austrian's syndrome) in 1881. Although the triad is now infrequent it still occurs.
- In section 2.3, change: “tariff” to “cost”.
- In the discussion, paragraph 3: $1,860 million would be easier to understand if written as $1.86 billion. Likewise, $1,800 million should be changed to $1.8 billion. You may also wish to express these costs in euros. $1.86 billion is ~ €1.6 billion and $1.8 billion is ~ €1.55 billion.
- In the conclusions, change the last sentence to: “The results of this study confirm the need to reinforce vaccination programs in adults at risk, especially those with comorbidities”.
Major comments:
- At the end of page 1 and beginning of page 2, the decreased incidence of invasive pneumococcal infections has been described by others as “herd immunity”. Your data suggests that reference 8 is inaccurate and that there is little evidence of herd immunity. In the discussion you mention “unsatisfactory vaccination coverages”. Do you mean that a suboptimal number of these individuals have been vaccinated? If so, say so and provide a percentage or percentages.
- The last paragraph of the introduction and the first paragraph of section 2.1 are nearly identical. However, otitis media is mentioned in the former and not the latter. S. pneumoniae can infect the lungs (pneumonia) or ears (otitis media), but it is considered “invasive” when it is found in the blood, spinal fluid (e.g., meningitis), or other site that normally does not have bacteria present. The difference between these paragraphs is confusing. Please clarify.
- In section 2.1 you have written “The CCDWH records data gathered from multiple Medicare data sources…”. Italy’s National Health Service automatically covers all citizens and legal foreign residents. Medicare is a national health insurance program in the United States, begun in 1965 under the Social Security Administration (SSA) and now administered by the Centers for Medicare and Medicaid Services (CMS). It primarily provides health insurance for Americans aged 65 and older, but also for some younger people with disability status as determined by the SSA, including people with end stage renal disease and amyotrophic lateral sclerosis. It seems highly unlikely that Medicare would have data from Liguria. Is your use of the word “Medicare” intended to be a brief substitute for Italy’s National Health Service? Please clarify.
- CMS required medical practices and Revenue Cycle Management (RCM) companies to make the switch from ICD-9 to ICD-10 by October 1, 2015, the last day for ICD-9 being September 30, 2015. Your data is from 2012-2018. The transition to ICD-10 enabled greater specificity in identifying health conditions. It also provided better data for measuring and tracking health care utilization and the quality of patient care. Table 1 suggests you exclusively used ICD-9 codes. The decision to use the old coding system suggests that inaccuracies were introduced into your data. Please discuss the reasons for the decision to use ICD-9 codes after 2015 and the potential impact of this decision on the accuracy of your data.
- In the final paragraph of the discussion, change: “As such, it provided both a highly specific estimate of cost, with a high degree of certainty that episodes included in the analysis were caused by S. pneumoniae, and one that is more sensitive, with the potential to capture additional undetected pneumococcal episodes and those potentially caused by other pathogens” to “As such, it provided both an accurate estimate of cost, with a high degree of certainty that episodes included in the analysis were caused by S. pneumoniae. It also provided sufficient sensitivity, with the potential to capture additional undetected pneumococcal episodes as well as those potentially caused by other pathogens”.
Author Response
The manuscript is generally well written. Attention to the following should improve it.
Minor comments:
1. How should one differentiate between sepsis, bacteremia, and bacteremic pneumonia? Please clarify.
-We clarified the case definitions as requested, in the section introduction.
2. The introduction would have more impact if you added the following:
a) Overall mortality from pneumococcal bacteremia ranges between 15 to 20 % in the antibiotic era. Pneumococcal endocarditis is rare but usually affects one or both left-sided valves (aortic > mitral). It’s important, because mortality remains high, ranging from 28–60%. Surgery may be required before a course of antibiotics is completed.
b) Osler described the clinical triad of pneumococcal endocarditis, meningitis, and pneumonia (AKA Austrian's syndrome) in 1881. Although the triad is now infrequent it still occurs.
-Thank you for your suggestions, we added this part as requested in the introduction section.
3. In section 2.3, change: “tariff” to “cost”.
-We modified this term as requested.
4. In the discussion, paragraph 3: $1,860 million would be easier to understand if written as $1.86 billion. Likewise, $1,800 million should be changed to $1.8 billion. You may also wish to express these costs in euros. $1.86 billion is ~ €1.6 billion and $1.8 billion is ~ €1.55 billion.
-We modified the costs as requested.
5. In the conclusions, change the last sentence to: “The results of this study confirm the need to reinforce vaccination programs in adults at risk, especially those with comorbidities”.
-We modified this sentence as requested.
Major comments:
1. At the end of page 1 and beginning of page 2, the decreased incidence of invasive pneumococcal infections has been described by others as “herd immunity”. Your data suggests that reference 8 is inaccurate and that there is little evidence of herd immunity. In the discussion you mention “unsatisfactory vaccination coverages”. Do you mean that a suboptimal number of these individuals have been vaccinated? If so, say so and provide a percentage or percentages.
-As suggested we replaced the incorrect reference into Van de Garde, M.D.B.; Knol, M.J.; Rots, N.Y.; Van Baarle, D.; Van Els, C.A.C.M. Vaccines to Protect Older Adults against Pneumococcal Disease. Interdiscip Top Gerontol Geriatr 2020, 43:113-130. doi: 10.1159/000504490.
2. The last paragraph of the introduction and the first paragraph of section 2.1 are nearly identical. However, otitis media is mentioned in the former and not the latter. S. pneumoniae can infect the lungs (pneumonia) or ears (otitis media), but it is considered “invasive” when it is found in the blood, spinal fluid (e.g., meningitis), or other site that normally does not have bacteria present. The difference between these paragraphs is confusing. Please clarify.
-Thank you, we modified the text as requested
3. In section 2.1 you have written “The CCDWH records data gathered from multiple Medicare data sources…”. Italy’s National Health Service automatically covers all citizens and legal foreign residents. Medicare is a national health insurance program in the United States, begun in 1965 under the Social Security Administration (SSA) and now administered by the Centers for Medicare and Medicaid Services (CMS). It primarily provides health insurance for Americans aged 65 and older, but also for some younger people with disability status as determined by the SSA, including people with end stage renal disease and amyotrophic lateral sclerosis. It seems highly unlikely that Medicare would have data from Liguria. Is your use of the word “Medicare” intended to be a brief substitute for Italy’s National Health Service? Please clarify.
-We agree with your comment and we modified the text to avoid misunderstandings accordingly.
4. CMS required medical practices and Revenue Cycle Management (RCM) companies to make the switch from ICD-9 to ICD-10 by October 1, 2015, the last day for ICD-9 being September 30, 2015. Your data is from 2012-2018. The transition to ICD-10 enabled greater specificity in identifying health conditions. It also provided better data for measuring and tracking health care utilization and the quality of patient care. Table 1 suggests you exclusively used ICD-9 codes. The decision to use the old coding system suggests that inaccuracies were introduced into your data. Please discuss the reasons for the decision to use ICD-9 codes after 2015 and the potential impact of this decision on the accuracy of your data.
-We thank you for you clarification, we explain the motivation on support of the ICD-9-CM codification use and we discussed this part in the discussion section.
The International Classification of Diseases 9th revision, Clinical Modifications (ICD‐9‐CM) is the version of ICD envisaged in Italy by the legislation for the coding of diseases and related problems.
The standard is now at its eleventh revision (ICD‐11), approved in 2018 by the Member States, at the 72 World Health Assembly and which will come into force on 1 January 2022.
ICD-9-CM was developed in 1977 and the last Italian version update dates back to 2007.
Thus, until now, in Italy, ICD-9-CM is the de facto standard for statistical studies on morbidity and mortality rates and for epidemiological studies, as well as a it is recognized as a valid management tool for public health and hygiene [https://www.fascicolosanitario.gov.it/en/ICD-9-CM. Last update: 05/08/2021].
5. In the final paragraph of the discussion, change: “As such, it provided both a highly specific estimate of cost, with a high degree of certainty that episodes included in the analysis were caused by S. pneumoniae, and one that is more sensitive, with the potential to capture additional undetected pneumococcal episodes and those potentially caused by other pathogens” to “As such, it provided both an accurate estimate of cost, with a high degree of certainty that episodes included in the analysis were caused by S. pneumoniae. It also provided sufficient sensitivity, with the potential to capture additional undetected pneumococcal episodes as well as those potentially caused by other pathogens
-We modified the main text as requested.
Round 2
Reviewer 1 Report
No answer given.
Reviewer 2 Report
The authors have satisfactorily addressed all the questions raised by this reviewer.